# Unsupervised Sequence Classification using Sequential Output Statistics

**Yu Liu** [†], **Jianshu Chen** [*], **and Li Deng**[†]

[*] Microsoft Research, Redmond, WA 98052, USA[*]
`jianshuc@microsoft.com`
[†] Citadel LLC, Seattle/Chicago, USA[†]
`Li.Deng@citadel.com`

## Abstract

We consider learning a sequence classifier without labeled data by using sequential output statistics. The problem is highly valuable since obtaining labels in training data is often costly, while the sequential output statistics (e.g., language models) could be obtained independently of input data and thus with low or no cost. To address the problem, we propose an unsupervised learning cost function and study its properties. We show that, compared to earlier works, it is less inclined to be stuck in trivial solutions and avoids the need for a strong generative model. Although it is harder to optimize in its functional form, a stochastic primal-dual gradient method is developed to effectively solve the problem. Experiment results on real-world datasets demonstrate that the new unsupervised learning method gives drastically lower errors than other baseline methods. Specifically, it reaches test errors about twice of those obtained by fully supervised learning.

## 1 Introduction

Unsupervised learning is one of the most challenging problems in machine learning. It is often formulated as the modeling of how the world works without requiring a huge amount of human labeling effort, e.g. [8]. To reach this grand goal, it is necessary to first solve a sub-goal of unsupervised learning with high practical value; that is, learning to predict output labels from input data without requiring costly labeled data. Toward this end, we study in this paper the learning of a sequence classifier without labels by using sequential output statistics. The problem is highly valuable since the sequential output statistics, such as language models, could be obtained independently of the input data and thus with no labeling cost.

The problem we consider here is different from most studies on unsupervised learning, which concern automatic discovery of inherent regularities of the input data to learn their representations [13, 28, 18, 17, 5, 1, 31, 20, 14, 12]. When these methods are applied in prediction tasks, either the learned representations are used as feature vectors [22] or the learned unsupervised models are used to initialize a supervised learning algorithm [9, 18, 2, 24, 10]. In both ways, the above unsupervised methods played an auxiliary role in helping supervised learning when it is applied to prediction tasks.

Recently, various solutions have been proposed to address the input-to-output prediction problem without using labeled training data, all without demonstrated successes [11, 30, 7]. Similar to this work, the authors in [7] proposed an unsupervised cost that also exploits the sequence prior of the output samples to train classifiers. The power of such a strong prior in the form of language

---

[*] All the three authors contributed equally to the paper.

[†] The work was done while Yu Liu and Li Deng were at Microsoft Research.

models in unsupervised learning was also demonstrated in earlier studies in [21, 3]. However, these earlier methods did not perform well in practical prediction tasks with real-world data without using additional strong generative models. Possible reasons are inappropriately formulated cost functions and inappropriate choices of optimization methods. For example, it was shown in [7] that optimizing the highly non-convex unsupervised cost function could easily get stuck in trivial solutions, although adding a special regularization mitigated the problem somewhat.

The solution provided in this paper fundamentally improves these prior works in [11, 30, 7] in following aspects. First, we propose a novel cost function for unsupervised learning, and find that it has a desired coverage-seeking property that makes the learning algorithm less inclined to be stuck in trivial solutions than the cost function in [7]. Second, we develop a special empirical formulation of this cost function that avoids the need for a strong generative model as in [30, 11].[3] Third, although the proposed cost function is more difficult to optimize in its functional form, we develop a stochastic primal-dual gradient (SPDG) algorithm to effectively solve problem. Our analysis of SPDG demonstrates how it is able to reduce the high barriers in the cost function by transforming it into a primal-dual domain. Finally and most importantly, we demonstrate the new cost function and the associated SPDG optimization algorithm work well in two real-world classification tasks. In the rest of the paper, we proceed to demonstrate these points and discuss related works along the way.

## 2 Empirical-ODM: An unsupervised learning cost for sequence classifiers

In this section, we extend the earlier work of [30] and propose an unsupervised learning cost named Empirical Output Distribution Match (Empirical-ODM) for training classifiers without labeled data. We first formulate the unsupervised learning problem with sequential output structures. Then, we introduce the Empirical-ODM cost and discuss its important properties that are closely related to unsupervised learning.

### 2.1 Problem formulation

We consider the problem of learning a sequence classifier that predicts an output sequence $(y_1, \ldots, y_{T_0})$ from an input sequence $(x_1, \ldots, x_{T_0})$ without using labeled data, where $T_0$ denotes the length of the sequence. Specifically, the learning algorithm does not have access to a labeled training set $\mathcal{D}_{XY} \triangleq \{(x_1^n, \ldots, x_{T_n}^n), (y_1^n, \ldots, y_{T_n}^n) : n = 1, \ldots, M\}$, where $T_n$ denotes the length of the $n$-th sequence. Instead, what is available is a collection of input sequences, denoted as $\mathcal{D}_X \triangleq \{(x_1^n, \ldots, x_{T_n}^n) : n = 1, \ldots, M\}$. In addition, we assume that the sequential output statistics (or sequence prior), in the form of an $N$-gram probability, are available:

$$p_{\text{LM}}(i_1, \ldots, i_N) \triangleq p_{\text{LM}}(y_{t-N+1}^n = i_1, \ldots, y_t^n = i_N)$$

where $i_1, \ldots, i_N \in \{1, \ldots, C\}$ and the subscript "LM" stands for language model. Our objective is to train the sequence classifier by just using $\mathcal{D}_X$ and $p_{\text{LM}}(\cdot)$. Note that the sequence prior $p_{\text{LM}}(\cdot)$, in the form of language models, is a type of structure commonly found in natural language data, which can be learned from a large amount of text data freely available without labeling cost. For example, in optical character recognition (OCR) tasks, $y_t^n$ could be an English character and $x_t^n$ is the input image containing this character. We can estimate an $N$-gram character-level language model $p_{\text{LM}}(\cdot)$ from a separate text corpus. Therefore, our learning algorithm will work in a fully unsupervised manner, without any human labeling cost. In our experiment section, we will demonstrate the effectiveness of our method on such a real OCR task. Other potential applications include speech recognition, machine translation, and image/video captioning.

In this paper, we focus on the sequence classifier in the form of $p_\theta(y_t^n|x_t^n)$ that is, it computes the posterior probability $p_\theta(y_t^n|x_t^n)$ only based on the current input sample $x_t^n$ in the sequence. Furthermore, we restrict our choice of $p_\theta(y_t^n|x_t^n)$ to be linear classifiers[4] and focus our attention on designing and understanding unsupervised learning costs and methods for label-free prediction. In

fact, as we will show in later sections, even with linear models, the unsupervised learning problem is still highly nontrivial and the cost function is also highly non-convex. And we emphasize that developing a successful unsupervised learning approach for linear classifiers, as we do in this paper, provides important insights and is an important first step towards more advanced nonlinear models (e.g., deep neural networks). We expect that, in future work, the insights obtained here could help us generalize our techniques to nonlinear models.

A recent work that shares the same motivations as our work is [29], which also recognizes the high cost of obtaining labeled data and seeks label-free prediction. Different from our setting, they exploit domain knowledge from laws of physics in computer vision applications, whereas our approach exploits sequential statistics in the natural language outputs. Finally, our problem is fundamentally different from the sequence transduction method in [15], although it also exploits language models for sequence prediction. Specifically, the method in [15] is a fully supervised learning in that it requires supervision at the sequence level; that is, for each input sequence, a corresponding output sequence (of possibly different length) is provided as a label. The use of language model in [15] only serves the purpose of regularization in the sequence-level *supervised* learning. In stark contrast, the unsupervised learning we propose does not require supervision at any level including specifically the sequence level; we do not need the sequence labels but only the prior distribution $p_{\mathrm{LM}}(\cdot)$ of the output sequences.

## 2.2 The Empirical-ODM

We now introduce an unsupervised learning cost that exploits the sequence structure in $p_{\mathrm{LM}}(\cdot)$. It is mainly inspired by the approach to breaking the Caesar cipher, one of the simplest forms of encryption [23]. Caesar cipher is a substitution cipher where each letter in the original message is replaced with a letter corresponding to a certain number of letters up or down in the alphabet. For example, the letter "D" is replaced by the letter "A", the letter "E" is replaced by the letter "B", and so on. In this way, the original message that was readable ends up being less understandable. The amount of this shifting is also known to the intended receiver of the message, who can decode the message by shifting back each letter in the encrypted message. However, Caesar cipher could also be broken by an unintended receiver (not knowing the shift) when it analyzes the frequencies of the letters in the encrypted messages and matches them up with the letter distribution of the original text [4, pp.9-11]. More formally, let $y_t = f(x_t)$ denote a function that maps each encrypted letter $x_t$ into an original letter $y_t$. And let $p_{\mathrm{LM}}(i) \triangleq p_{\mathrm{LM}}(y_t = i)$ denote the prior letter distribution of the original message, estimated from a regular text corpus. When $f(\cdot)$ is constructed in a way that all mapped letters $\{y_t : y_t = f(x_t), t = 1, \ldots, T\}$ have the same distribution as the prior $p_{\mathrm{LM}}(i)$, it is able to break the Caesar cipher and recover the original letters at the mapping outputs.

Inspired by the above approach, the posterior probability $p_\theta(y_t^n | x_t^n)$ in our classification problem can be interpreted as a stochastic mapping, which maps each input vector $x_t^n$ (the "encrypted letter") into an output vector $y_t^n$ (the "original letter") with probability $p_\theta(y_t^n | x_t^n)$. Then in a samplewise manner, each input sequence $(x_1^n, \ldots, x_{T_n}^n)$ is stochastically mapped into an output sequence $(y_1^n, \ldots, y_{T_n}^n)$. We move a step further than the above approach by requiring that the distribution of the $N$-grams among all the mapped output sequences are close to the prior $N$-gram distribution $p_{\mathrm{LM}}(i_1, \ldots, i_N)$. With this motivation, we propose to learn the classifier $p_\theta(y_t | x_t)$ by minimizing the cross entropy between the prior distribution and the expected $N$-gram frequency of the output sequences:

$$\min_\theta \left\{ \mathcal{J}(\theta) \triangleq - \sum_{i_1, \ldots, i_N} p_{\mathrm{LM}}(i_1, \ldots, i_N) \ln \overline{p}_\theta(i_1, \ldots, i_N) \right\} \tag{1}$$

where $\overline{p}_\theta(i_1, \ldots, i_N)$ denotes the expected frequency of a given $N$-gram $(i_1, \ldots, i_N)$ among all the output sequences. In Appendix B of the supplementary material, we derive its expression as

$$\overline{p}_\theta(i_1, \ldots, i_N) \triangleq \frac{1}{T} \sum_{n=1}^{M} \sum_{t=1}^{T_n} \prod_{k=0}^{N-1} p_\theta(y_{t-k}^n = i_{N-k} | x_{t-k}^n) \tag{2}$$

where $T \triangleq T_1 + \cdots + T_M$ is the total number of samples in all sequences. Note that minimizing the cross entropy in (1) is also equivalent to minimizing the Kullback-Leibler (KL) divergence between the two distributions since they only differ by a constant term, $\sum p_{\mathrm{LM}} \ln p_{\mathrm{LM}}$. Therefore, the cost function (1) seeks to estimate $\theta$ by matching the two output distributions, where the expected $N$-gram

distribution in (2) is an empirical average over all the samples in the training set. For this reason, we name the cost (1) as *Empirical Output Distribution Match* (Empirical-ODM) cost.

In [30], the authors proposed to minimize an output distribution match (ODM) cost, defined as the KL-divergence between the prior output distribution and the marginalized output distribution, $D(p_{\text{LM}}(y)||p_\theta(y))$, where $p_\theta(y) \triangleq \int p_\theta(y|x)p(x)dx$. However, evaluating $p_\theta(y)$ requires integrating over the input space using a generative model $p(x)$. Due to the lack of such a generative model, they were not able to optimize this proposed ODM cost. Instead, alternative approaches such as Dual autoencoders and GANs were proposed as heuristics. Their results were not successful without using a few labeled data. Our proposed Empirical-ODM cost is different from the ODM cost in [30] in three key aspects. (i) We do not need any labeled data for training. (ii) We exploit sequence structure of output statistics, i.e., in our case $y = (y_1, \ldots, y_N)$ ($N$-gram) whereas in [30] $y = y_t$ (unigram, i.e., no sequence structure). This is crucial in developing a working unsupervised learning algorithm. The change from unigram to $N$-gram allows us to explicitly exploit the sequence structures at the output, which makes the technique from non-working to working (see Table 2 in Section 4). It might also explain why the method in [30] failed as it does not exploit the sequence structure. (iii) We replace the marginalized distribution $p_\theta(y)$ by the expected $N$-gram frequency in (2). This is critical in that it allows us to directly minimize the divergence between two output distributions without the need for a generative model, which [30] could not do. In fact, we can further show that $\overline{p}_\theta(i_1, \ldots, i_N)$ is an empirical approximation of $p_\theta(y)$ with $y = (y_1, \ldots, y_N)$ (see Appendix B.2 of the supplementary material). In this way, our cost (1) can be understood as an $N$-*gram* and *empirical* version of the ODM cost except for an additive constant, i.e., $y$ is replaced by $y = (y_1, \ldots, y_N)$ and $p_\theta(y)$ is replaced by its empirical approximation.

## 2.3 Coverage-seeking versus mode-seeking

We now discuss an important property of the proposed Empirical-ODM cost (1) by comparing it with the cost proposed in [7]. We show that the Empirical-ODM cost has a *coverage-seeking* property, which makes it more suitable for unsupervised learning than the *mode-seeking* cost in [7].

In [7], the authors proposed the expected negative log-likelihood as the unsupervised learning cost function that exploits the output sequential statistics. The intuition was to maximize the aggregated log-likelihood of all the output sequences assumed to be generated by the stochastic mapping $p_\theta(y_t^n|x_t^n)$. We show in Appendix A of the supplementary material that their cost is equivalent to

$$- \sum_{i_1, \ldots, i_{N-1}} \sum_{i_N} \overline{p}_\theta(i_1, \ldots, i_N) \ln p_{\text{LM}}(i_N|i_{N-1}, \ldots, i_1) \tag{3}$$

where $p_{\text{LM}}(i_N|i_{N-1}, \ldots, i_1) \triangleq p(y_t^n = i_N|y_{t-1}^n = i_{N-1}, \ldots, y_{t-N+1}^n = i_1)$, and the summations are over all possible values of $i_1, \ldots, i_N \in \{1, \ldots, C\}$. In contrast, we can rewrite our cost (1) as

$$- \sum_{i_1, \ldots, i_{N-1}} p_{\text{LM}}(i_1, \ldots, i_{N-1}) \cdot \sum_{i_N} p_{\text{LM}}(i_N|i_{N-1}, \ldots, i_1) \ln \overline{p}_\theta(i_1, \ldots, i_N) \tag{4}$$

where we used the chain rule of conditional probabilities. Note that both costs (3) and (4) are in a cross entropy form. However, a key difference is that the positions of the distributions $\overline{p}_\theta(\cdot)$ and $p_{\text{LM}}(\cdot)$ are swapped. We show that the cost in the form of (3) proposed in [7] is a *mode-seeking* divergence between two distributions, while by swapping $\overline{p}_\theta(\cdot)$ and $p_{\text{LM}}(\cdot)$, our cost in (4) becomes a *coverage-seeking* divergence (see [25] for a detailed discussion on divergences with these two different behaviors). To understand this, we consider the following two situations:

- If $p_{\text{LM}}(i_N|i_{N-1}, \ldots, i_1) \to 0$ and $\overline{p}_\theta(i_1, \ldots, i_N) > 0$ for a certain $(i_1, \ldots, i_N)$, the cross entropy in (3) goes to $+\infty$ and the cross entropy in (4) approaches zero.
- If $p_{\text{LM}}(i_N|i_{N-1}, \ldots, i_1) > 0$ and $\overline{p}_\theta(i_1, \ldots, i_N) \to 0$ for a certain $(i_1, \ldots, i_N)$, the cross entropy in (3) approaches zero and the cross entropy in (4) goes to $+\infty$.

Therefore, the cost function (3) will heavily penalize the classifier if it predicts an output that is believed to be less probable by the prior distribution $p_{\text{LM}}(\cdot)$, and it will not penalize the classifier when it does not predict an output that $p_{\text{LM}}(\cdot)$ believes to be probable. That is, the classifier is encouraged to predict a single output mode with high probability in $p_{\text{LM}}(\cdot)$, a behavior called "mode-seeking" in [25]. This probably explains the phenomena observed in [7]: the training process easily converges to

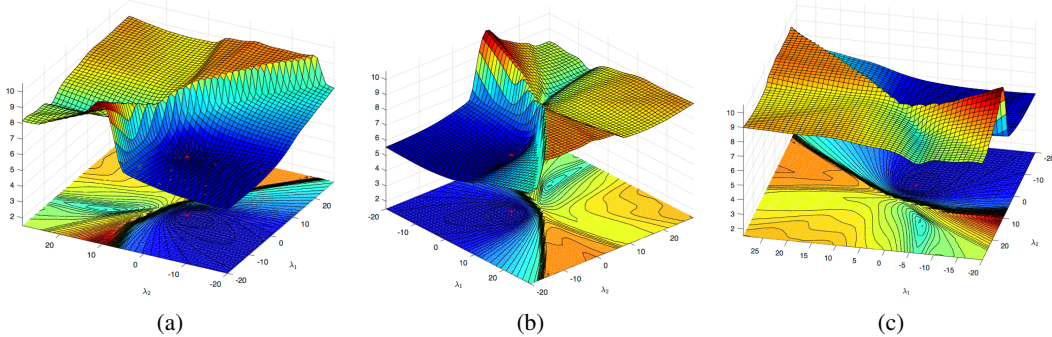

<div align="center">(a)           (b)           (c)</div>

Figure 1: The profiles of $\mathcal{J}(\theta)$ for the OCR dataset on a two-dimensional affine space passing through the supervised solution. The three figures show the same profile from different angles, where the red dot is the supervised solution. The contours of the profiles are shown at the bottom.

a trivial solution of predicting the same output that has the largest probability in $p_{\text{LM}}(\cdot)$. In contrast, the cost (4) will heavily penalize the classifier if it does not predict the output for which $p_{\text{LM}}(\cdot)$ is positive, and will penalize less if it predicts outputs for which $p_{\text{LM}}(\cdot)$ is zero. That is, this cost will encourage $p_\theta(y|x)$ to cover as much of $p_{\text{LM}}(\cdot)$ as possible, a behavior called "coverage-seeking" in [25]. Therefore, training the classifier using (4) will make it less inclined to learn trivial solutions than that in [7] since it will be heavily penalized. We will verify this fact in our experiment section 4. In addition, the coverage-seeking property could make the learning less sensitive to the sparseness of language models (i.e., $p_{\text{LM}}$ is zero for some $N$-grams) since the cost will not penalize these $N$-grams. In summary, our proposed cost (1) is more suitable for unsupervised learning than that in [7].

### 2.4 The difficulties of optimizing $\mathcal{J}(\theta)$

However, there are two main challenges of optimizing the Empirical-ODM cost $\mathcal{J}(\theta)$ in (1). The first one is that the sample average (over the entire training data set) in the expression of $\overline{p}_\theta(\cdot)$ (see (2)) is inside the logarithmic loss, which is different from traditional machine learning problems where the average is outside loss functions (e.g., $\sum_t f_t(\theta)$). This functional form prevents us from applying stochastic gradient descent (SGD) to minimize (1) as the stochastic gradients would be intrinsically biased (see Appendix C for a detailed discussion and see section 4 for the experiment results). The second challenge is that the cost function $\mathcal{J}(\theta)$ is highly non-convex even with linear classifiers. To see this, we visualize the profile of the cost function $\mathcal{J}(\theta)$ (restricted to a two-dimensional sub-space) around the supervised solution in Figure 1.[5][6] We observe that there are local optimal solutions and there are high barriers between the local and global optimal solutions. Therefore, besides the difficulty of having the sample average inside the logarithmic loss, minimizing this cost function directly will be difficult since crossing the high barriers to reach the global optimal solution would be hard if not properly initialized.

## 3 The Stochastic Primal-Dual Gradient (SPDG) Algorithm

To address the first difficulty in Section 2.4, we transform the original cost (1) into an equivalent min-max problem in order to bring the sample average out of the logarithmic loss. Then, we could obtain unbiased stochastic gradients to solve the problem. To this end, we first introduce the concept of *convex conjugate functions*. For a given convex function $f(u)$, its convex conjugate function $f^\star(\nu)$ is defined as $f^\star(\nu) \triangleq \sup_u (\nu^T u - f(u))$ [6, pp.90-95], where $u$ and $\nu$ are called primal and dual variables, respectively. For a scalar function $f(u) = -\ln u$, its conjugate function can be calculated as $f^\star(\nu) = -1 - \ln(-\nu)$ with $\nu < 0$. Furthermore, it holds that $f(u) = \sup_\nu (u^T \nu - f^\star(\nu))$, by

**Algorithm 1** Stochastic Primal-Dual Gradient Method

---

1: **Input data:** $\mathcal{D}_X = \{(x_1^n, \ldots, x_{T_n}^n) : n = 1, \ldots, M\}$ and $p_{\text{LM}}(i_1, \ldots, i_N)$.
2: Initialize $\theta$ and $V$ where the elements of $V$ are negative
3: **repeat**
4:     Randomly sample a mini-batch of $B$ subsequences of length $N$ from all the sequences in the training set $\mathcal{D}_X$, i.e., $\mathcal{B} = \{(x_{t_m-N+1}^{n_m}, \ldots, x_{t_m}^{n_m})\}_{m=1}^B$.
5:     Compute the stochastic gradients for each subsequence in the mini-batch and average them

$$\Delta\theta = \frac{1}{B}\sum_{m=1}^B \frac{\partial L_{t_m}^{n_m}}{\partial\theta}, \quad \Delta V = \frac{1}{B}\sum_{m=1}^B \frac{\partial L_{t_m}^{n_m}}{\partial V} + \frac{\partial}{\partial V}\sum_{i_1 \cdots i_N} p_{\text{LM}}(i_1, \ldots, i_N)\ln(-\nu_{i_1, \ldots, i_N})$$

6:     Update $\theta$ and $V$ according to $\theta \leftarrow \theta - \mu_\theta\Delta\theta$ and $V \leftarrow V + \mu_v\Delta V$.
7: **until** convergence or a certain stopping condition is met

---

which we have $-\ln u = \max_\nu(u\nu + 1 + \ln(-\nu))$.[7] Substituting it into (1), the original minimization problem becomes the following equivalent min-max problem:

$$\min_\theta \max_{\{\nu_{i_1, \ldots, i_N} < 0\}} \left\{ \mathcal{L}(\theta, V) \triangleq \frac{1}{T}\sum_{n=1}^M \sum_{t=1}^{T_n} L_t^n(\theta, V) + \sum_{i_1, \ldots, i_N} p_{\text{LM}}(i_1, \ldots, i_N)\ln(-\nu_{i_1, \ldots, i_N}) \right\} \quad (5)$$

where $V \triangleq \{\nu_{i_1, \ldots, i_N}\}$ is a collection of all the dual variables $\nu_{i_1, \ldots, i_N}$, and $L_t^n(\theta, V)$ is the $t$-th component function in the $n$-th sequence, defined as

$$L_t^n(\theta, V) \triangleq \sum_{i_1, \ldots, i_N} p_{\text{LM}}(i_1, \ldots, i_N)\nu_{i_1, \ldots, i_N} \prod_{k=0}^{N-1} p_\theta(y_{t-k}^n = i_{N-k}|x_{t-k}^n)$$

In the equivalent min-max problem (5), we find the optimal solution $(\theta^\star, V^\star)$ by minimizing $\mathcal{L}$ with respect to the *primal* variable $\theta$ and maximizing $\mathcal{L}$ with respect to the *dual* variable $V$. The obtained optimal solution to (5), $(\theta^\star, V^\star)$, is called the saddle point of $\mathcal{L}$ [6]. Once it is obtained, we only keep $\theta^\star$, which is also the optimal solution to (1) and thus the model parameter.

We further note that the equivalent min-max problem (5) is now in a form that sums over $T = T_1 + \cdots + T_M$ component functions $L_t^n(\theta, V)$. Therefore, the empirical average has been brought out of the logarithmic loss and we are ready to apply stochastic gradient methods. Specifically, we minimize $\mathcal{L}$ with respect to the *primal* variable $\theta$ by stochastic gradient *descent* and maximize $\mathcal{L}$ with respect to the *dual* variable $V$ by stochastic gradient *ascent*. Therefore, we name the algorithm stochastic primal-dual gradient (SPDG) method (see its details in Algorithm 1). We implement the SPDG algorithm in TensorFlow, which automatically computes the stochastic gradients.[8] Finally, the constraint on dual variables $\nu_{i_1, \ldots, i_N}$ are automatically enforced by the inherent log-barrier, $\ln(-\nu_{i_1, \ldots, i_N})$, in (5) [6]. Therefore, we do not need a separate method to enforce the constraint.

We now show that the above min-max (primal-dual) reformulation also alleviates the second difficulty discussed in Section 2.4. Similar to the case of $\mathcal{J}(\theta)$, we examine the profile of $\mathcal{L}(\theta, V)$ in (5) (restricted to a two-dimensional sub-space) around the optimal (supervised) solution in Figure 2a (see Appendix F for the visualization details). Comparing Figure 2a to Figure 1, we observe that the profile of $\mathcal{L}(\theta, V)$ is smoother than that of $\mathcal{J}(\theta)$ and the barrier is significantly lower. To further compare $\mathcal{J}(\theta)$ and $\mathcal{L}(\theta, V)$, we plot in Figure 2b the values of $\mathcal{J}(\theta)$ and $\mathcal{L}(\theta, V)$ along the same line of $\theta^\star + \lambda_p(\theta_1 - \theta^\star)$ for different $\lambda_p$. It shows that the barrier of $\mathcal{L}(\theta, V)$ along the primal direction is lower than that in $\mathcal{J}(\theta)$. These observations imply that the reformulated min-max problem (5) is better conditioned than the original problem (1), which further justifies the use of SPDG method.

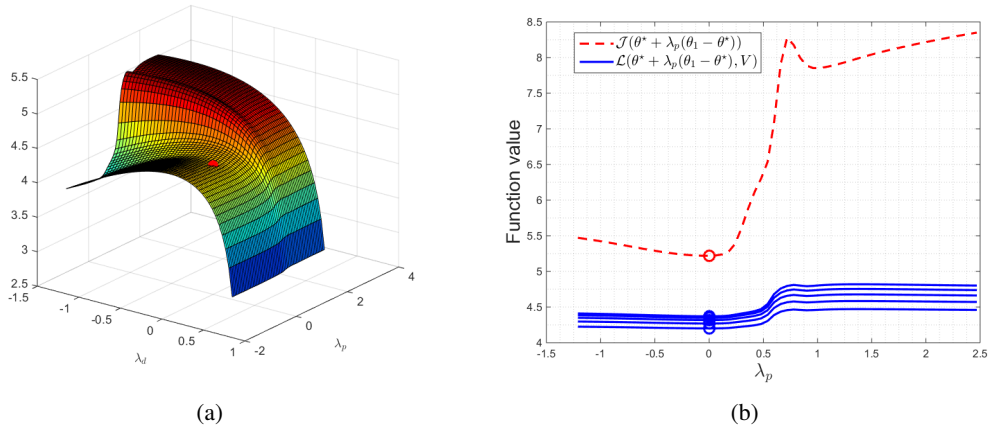

(a)                                                    (b)

Figure 2: The profiles of $\mathcal{L}(\theta, V)$ for the OCR dataset. (a) The profile on a two-dimensional affine space passing through the optimal solution (red dot). (b) The profile along the line of $\theta^\star + \lambda_p(\theta_1 - \theta^\star)$ for different values of $\lambda_p \in \mathbb{R}$, where the circles are the optimal solutions.

## 4 Experiments

### 4.1 Experimental setup

We evaluate our unsupervised learning scheme described in earlier seciton using two classification tasks, unsupervised character-level OCR and unsupervised English Spelling Correction (Spell-Corr). In both tasks, there is no label provided during training. Hence, they are both unsupervised.

For the OCR task, we obtain our dataset from a public database UWIII English Document Image Database [27], which contains images for each line of text with its corresponding groudtruth. We first use Tesseract [19] to segment the image for each line of text into characters tiles and assign each tile with one character. We verify the segmentation result by training a simple neural network classifier on the segmented results and achieve 0.9% error rate on the test set. Then, we select sentence segments that are longer than 100 and contain only lowercase English characters and common punctuations (space, comma, and period). As a result, we have a vocabulary of size 29 and we obtain 1,175 sentence segments including 153,221 characters for our OCR task. To represent images, we extract VGG19 features with $dim = 4096$, and project them into 200-dimension vectors using Principal Component Analysis. We train the language models (LM) $p_{\mathrm{LM}}(\cdot)$ to provide the required output sequence statistics from both in-domain and out-of-domain data sources. The out-of-domain data sources are completely different databases, including three different language partitions (CNA, NYT, XIN) in the English Gigaword database [26].

In Spell-Corr task, we learn to correct the spelling from a mis-spelled text. From the AFP partition of the Gigaword database, we select 500 sentence segments into our Spell-Corr dataset. We select sentences that are longer than 100 and contain only English characters and common punctuations, resulting in a total of 83,567 characters. The mis-spelled texts are generated by substitution simulations and are treated as our inputs. The objective of this task is to recover the original text.

### 4.2 Results: Comparing optimization algorithms

In the first set of experiments, we aim to evaluate the effectiveness of the SPDG method as described in Section 3, which is designed for optimizing the Empirical-ODM cost in Section 2. The analysis provided in Sections 2 and 3 sheds insight to why SPDG is superior to the method in [7] and to the standard stochastic gradient descent (SGD) method. The coverage-seeking behavior of the proposed Empirical-ODM cost helps avoid trivial solutions, and the simultaneous optimization of primal-dual variables reduces the barriers in the highly non-convex profile of the cost function. Furthermore, we do not include the methods from [30] because their approaches could not achieve satisfactory results without a few labeled data, while we only consider fully unsupervised learning setting. In addition, the methods in [30] are not optimizing the ODM cost and do not exploit the output sequential statistics.

Table 1 provides strong experimental evidence demonstrating the substantially greater effectiveness of the primal-dual method over the SGD and the method in [7] on both tasks. All these results are obtained by training the models until converge. Let us examine the results on the OCR in detail. First, the SPDG on the unsupervised cost function achieves 9.21% error rate, much lower than the error rates of any of mini-batch SGD runs, where the size of the mini-batches ranges from 10 to 10,000. Note that, larger mini-batch sizes produce lower errors here because it becomes closer to full-batch gradient and thus lower bias in SGD. On the other hand, when the mini-batch size is as small as 10, the high error rate of 83.09% is close to a guess by majority rule — predicting the character (space) that has a largest proportion in the train set, i.e., $25,499/153,221 = 83.37\%$. Furthermore, the method from [7] does not perform well no matter how we tune the hyperparameters for the generative regularization. Finally and perhaps most interestingly, with no labels provided in the training, the classification errors produced by our method are only about twice compared with supervised learning (4.63% shown in Table 1). This clearly demonstrates that the unsupervised learning scheme proposed in this paper is an effective one. For the Spelling Correction data set (see the third column in Table 1), we observe rather consistent results with the OCR data set.

Table 1: Test error rates on two datasets: OCR and Spell-Corr. The 2-gram character LM is trained from in-domain data. The numbers inside $\langle \cdot \rangle$ are the mini-batch sizes of the SGD method.

| Data sets | SPDG (Ours) | Method from [7] | SGD $\langle 10 \rangle$ | SGD $\langle 100 \rangle$ | SGD $\langle 1k \rangle$ | SGD $\langle 10k \rangle$ | Supervised Learning | Majority Guess |
|---|---|---|---|---|---|---|---|---|
| OCR | **9.59%** | 83.37% | 83.09% | 78.05% | 67.14% | 56.48% | 4.63% | 83.37% |
| Spell-Corr | **1.94%** | 82.91% | 82.91% | 72.93% | 65.69% | 45.24% | 0.00% | 82.91% |

### 4.3 Results: Comparing orders of language modeling

In the second set of experiments, we examine to what extent the use of sequential statistics (e.g. 2- and 3-gram LMs) can do better than the uni-gram LM (no sequential information) in unsupervised learning. The unsupervised prediction results are shown in Table 2, using different data sources to estimate N-gram LM parameters. Consistent across all four ways of estimating reliable N-gram LMs, we observe significantly lower error rates when the unsupervised learning exploits 2-gram and 3-gram LM as sequential statistics compared with exploiting the prior with no sequential statistics (i.e. 1-gram). In three of four cases, exploiting a 3-gram LM gives better results than a 2-gram LM. Furthermore, the comparable error rate associated with 3-gram using out-of-domain output character data (10.17% in Table 2) to that using in-domain output character data (9.59% in Table 1) indicates that the effectiveness of the unsupervised learning paradigm presented in this paper is robust to the quality of the LM acting as the sequential prior.

Table 2: Test error rates on the OCR dataset. Character-level language models (LMs) with the orders are trained from three out-of-domain datasets and from the fused in-domain and out-of-domain data.

| | NYT-LM | XIN-LM | CNA-LM | Fused-LM |
|---|---|---|---|---|
| No. Sents | 1,206,903 | 155,647 | 12,234 | 15,409 |
| No. Chars | 86,005,542 | 18,626,451 | 1,911,124 | 2,064,345 |
| 1-gram | 71.83% | 72.14% | 71.51% | 71.25% |
| 2-gram | 10.93% | **12.55%** | 10.56% | 10.33% |
| 3-gram | **10.17%** | 12.89% | **10.29%** | **9.21%** |

## 5 Conclusions and future work

In this paper, we study the problem of learning a sequence classifier without the need for labeled training data. The practical benefit of such unsupervised learning is tremendous. For example, in large scale speech recognition systems, the currently dominant supervised learning methods typically require a few thousand hours of training data, where each utterance in the acoustic form needs to be labeled by humans. Although there are millions of hours of natural speech data available for training, labeling all of them for supervised learning is less feasible. To make effective use of such

huge amounts of acoustic data, the practical unsupervised learning approach discussed in this paper would be called for. Other potential applications such as machine translation, image and video captioning could also benefit from our paradigm. This is mainly because of their common natural language output structure, from which we could exploit the sequential structures for learning the classifier without labels. For other (non-natural-language) applications where there is also a sequential output strucutre, our proposed approach could be applicable in a similar manner. Furthermore, our proposed Empirical-ODM cost function significantly improves over the one in [7] by emphasizing the coverage-seeking behavior. Although the new cost function has a functional form that is more difficult to optimize, a novel SPDG algorithm is developed to effectively address the problem. An analysis of profiles of the cost functions sheds insight to why SPDG works well and why previous methods failed. Finally, we demonstrate in two datasets that our unsupervised learning method is highly effective, producing only about twice errors as fully supervised learning, which no previous unsupervised learning could produce without additional steps of supervised learning. While the current work is restricted to linear classifiers, we intend to generalize the approach to nonlinear models (e.g., deep neural nets [16]) in our future work. We also plan to extend our current method from exploiting N-gram LM to exploiting the currently state-of-the-art neural-LM. Finally, one challenge that remains to be addressed is the scaling of the current method to large vocabulary and high-order LM (i.e., large $C$ and $N$). In this case, the summation over all $(i_1, \ldots, i_N)$ in (5) becomes computationally expensive. A potential solution is to parameterize the dual variable $\nu_{i_1,\ldots,i_N}$ by a recurrent neural network and approximate the sum using beamsearch, which we leave as a future work.

**Acknowledgments**

The authors would like to thank all the anonymous reviewers for their constructive feedback.

## Footnotes

[3]The work [11] only proposed a conceptual idea of using generative models to integrate the output structure and the output-to-input structure for unsupervised learning in speech recognition. Specifically, the generative models are built from the domain knowledge of speech waveform generation mechanism. No mathematical formulation or successful experimental results are provided in [11].

[4]$p_\theta(y_t^n = i|x_t^n) = e^{\gamma w_i^T x_t^n} / \sum_{j=1}^C e^{\gamma w_j^T x_t^n}$, where the model parameter is $\theta \triangleq \{w_i \in \mathbb{R}^d, i = 1, \ldots, C\}$.

[5]The approach to visualizing the profile is explained with more detail in Appendix F. More slices and a video of the profiles from many angles can be found in the supplementary material.

[6]Note that the supervised solution (red dot) coincides with the global optimal solution of $\mathcal{J}(\theta)$. The intuition for this is that the classifier trained by supervised learning should also produce output $N$-gram distribution that is close to the prior marginal output $N$-gram distribution given by $p_{\text{LM}}(\cdot)$.

[7]The supremum is attainable and is thus replaced by maximum.

[8]The code will be released soon.

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
