[Supplementary Material]

# Supplementary Material for "Unsupervised Sequence Classification using Sequential Output Statistics"

## A  Derivation of the equivalent form of the cost in [7]

The cost function in [7] can be expressed as:

$$\mathbb{E}\Big[-\sum_{n=1}^{M}\ln p_{\mathrm{LM}}(y_1^n,\ldots,y_{T_n}^n)|x_1^n,\ldots,x_{T_n}^n\Big] \tag{6}$$

We now show how to derive (3) from the above expression. In $N$-gram case, the language model can be written as

$$p_{\mathrm{LM}}(y_1^n,\ldots,y_{T_n}^n)=\prod_{t=1}^{T_n}p_{\mathrm{LM}}(y_t^n|y_{t-1}^n,\ldots,y_{t-N+1}^n)$$

Substituting the above expression into the cost (6), we obtain

$$\mathbb{E}\Big[-\sum_{n=1}^{M}\ln p_{\mathrm{LM}}(y_1^n,\ldots,y_{T_n}^n)|x_1^n,\ldots,x_{T_n}^n\Big]$$

$$=-\sum_{n=1}^{M}\sum_{(y_1^n,\ldots,y_{T_n}^n)}\prod_{t=1}^{T_n}p_\theta(y_t^n|x_t^n)\ln p_{\mathrm{LM}}(y_1^n,\ldots,y_{T_n}^n)$$

$$=-\sum_{n=1}^{M}\sum_{(y_1^n,\ldots,y_{T_n}^n)}p_\theta(y_1^n|x_1^n)\cdots p_\theta(y_{T_n}^n|x_{T_n}^n)\times\sum_{t=1}^{T_n}\ln p_{\mathrm{LM}}(y_t^n|y_{t-1}^n,\ldots,y_{t-N+1}^n)$$

$$=-\sum_{n=1}^{M}\sum_{t=1}^{T_n}\sum_{(y_1^n,\ldots,y_{T_n}^n)}p_\theta(y_1^n|x_1^n)\cdots p_\theta(y_{T_n}^n|x_{T_n}^n)\times\ln p_{\mathrm{LM}}(y_t^n|y_{t-1}^n,\ldots,y_{t-N+1}^n)$$

$$=-\sum_{n=1}^{M}\sum_{t=1}^{T_n}\sum_{(y_t^n,\ldots,y_{t-N+1}^n)}p_\theta(y_t^n|x_t^n)\cdots p_\theta(y_{t-N+1}^n|x_{t-N+1}^n)\times\ln p_{\mathrm{LM}}(y_t^n|y_{t-1}^n,\ldots,y_{t-N+1}^n)$$

$$\times\sum_{y_1^n,\ldots,y_{t-N}^n}p_\theta(y_1^n|x_1^n)\cdots p_\theta(y_{t-N}^n|x_{t-N}^n)$$

$$\times\sum_{y_{t+1}^n,\ldots,y_{T_n}^n}p_\theta(y_{t+1}^n|x_{t+1}^n)\cdots p_\theta(y_{T_n}^n|x_{T_n}^n)$$

$$=-\sum_{n=1}^{M}\sum_{t=1}^{T_n}\sum_{(y_t^n,\ldots,y_{t-N+1}^n)}p_\theta(y_t^n|x_t^n)\cdots p_\theta(y_{t-N+1}^n|x_{t-N+1}^n)\times\ln p_{\mathrm{LM}}(y_t^n|y_{t-1}^n,\ldots,y_{t-N+1}^n)$$

$$=-\sum_{n=1}^{M}\sum_{t=1}^{T_n}\sum_{i_1,\ldots,i_N}p_\theta(y_t^n=i_N|x_t)\cdots p_\theta(y_{t-N+1}^n=i_1|x_{t-N+1}^n)$$

$$\times\ln p_{\mathrm{LM}}(y_t^n=i_N|y_{t-1}^n=i_{N-1},\ldots,y_{t-N+1}^n=i_1)$$

$$=-\sum_{n=1}^{M}\sum_{t=1}^{T_n}\sum_{i_1,\ldots,i_N}p_\theta(y_t^n=i_N|x_t^n)\cdots p_\theta(y_{t-N+1}^n=i_1|x_{t-N+1}^n)\times\ln p_{\mathrm{LM}}(i_N|i_{N-1},\ldots,i_1)$$

$$=-\sum_{i_1,\ldots,i_N}\ln p_{\mathrm{LM}}(i_N|i_{N-1},\ldots,i_1)\times\sum_{n=1}^{M}\sum_{t=1}^{T_n}p_\theta(y_t^n=i_N|x_t)\cdots p_\theta(y_{t-N+1}^n=i_1|x_{t-N+1}^n)$$

$$=-T\sum_{i_1,\ldots,i_N}\ln p_{\mathrm{LM}}(i_N|i_{N-1},\ldots,i_1)\times\frac{1}{T}\sum_{n=1}^{M}\sum_{t=1}^{T_n}p_\theta(y_t^n=i_N|x_t^n)\cdots p_\theta(y_{t-N+1}^n=i_1|x_{t-N+1}^n)$$

# B  Properties of $\bar{p}_\theta(i_1, \ldots, i_N)$

## B.1  $\bar{p}_\theta(i_1, \ldots, i_N)$ is the expected $N$-gram frequency of all the output sequences

In this section, we formally derive the following relation, which interprets $\bar{p}_\theta(i_1, \ldots, i_N)$ as the expected frequency of $(i_1, \ldots, i_N)$ in the output sequence:

$$\mathbb{E}_{\prod_{n=1}^M \prod_{t=1}^{T_n} p_\theta(y_t^n|x_t^n)} \left[ \frac{n(i_1, \ldots, i_N)}{T} \right] = \bar{p}_\theta(i_N, \ldots, i_1)$$

where $T \triangleq T_1 + \cdots T_M$. Let $(x_1^n, \ldots, x_{T_n}^n)$ be a given $n$-th input training sequence, and let $(y_1^n, \ldots, y_{T_n}^n)$ be a sequence generated according to the posterior $\prod_{t=1}^{T_n} p_\theta(y_t^n|x_t^n)$ (which is the classifier). Furthermore, let $\mathbb{I}_t^n(i_1, \ldots, i_N)$ denote the indicator function of the event $\{y_{t-N+1}^n = i_1, \ldots, y_t^n = i_N\}$, and let $n(i_1, \ldots, i_N)$ denote the number of the $N$-gram $(i_1, \ldots, i_N)$ appearing in all the output sequences $\{(y_1^n, \ldots, y_{T_n}^n) : n = 1, \ldots, M\}$. Then, we have the following relation:

$$n(i_1, \ldots, i_N) = \sum_{n=1}^M \sum_{t=1}^{T_n} \mathbb{I}_t^n(i_1, \ldots, i_N)$$

Obviously, $n(i_1, \ldots, i_N)$ is a function of $\{(y_1^n, \ldots, y_{T_n}^n) : n = 1, \ldots, M\}$ and is thus a random variable. Taking the conditional expectation of the above expression with respect to $\prod_{n=1}^M \prod_{t=1}^{T_n} p_\theta(y_t^n|x_t^n)$, we obtain

$$\mathbb{E}_{\prod_{n=1}^M \prod_{t=1}^{T_n} p_\theta(y_t^n|x_t^n)}[n(i_1, \ldots, i_N)]$$

$$= \sum_{n=1}^M \sum_{t=1}^{T_n} \mathbb{E}_{\prod_{n=1}^M \prod_{t=1}^{T_n} p_\theta(y_t^n|x_t^n)}[\mathbb{I}_t^n(i_1, \ldots, i_N)]$$

$$= \sum_{n=1}^M \sum_{t=1}^{T_n} \mathbb{E}_{\prod_{t=1}^{T_n} p_\theta(y_t^n|x_t^n)}[\mathbb{I}_t^n(i_1, \ldots, i_N)]$$

$$\overset{(a)}{=} \sum_{n=1}^M \sum_{t=1}^{T_n} \prod_{k=0}^{N-1} p_\theta(y_{t-k}^n = i_{N-k}|x_{t-k}^n)$$

where step (a) uses the fact that the expectation of an indicator function of an event equals the probability of the event. Divide both sides by $T$, the right hand side of the above expression becomes $\bar{p}_\theta(i_1, \ldots, i_N)$, and we conclude our proof.

## B.2  $\bar{p}_\theta(i_N, \ldots, i_1)$ is an empirical approximation of the marginal output $N$-gram probability

First, define the marginal $N$-gram probability $p_\theta(i_1, \ldots, i_N)$ as

$$p_\theta(i_1, \ldots, i_N) \triangleq p_\theta(y_1 = i_1, \ldots, y_N = i_N) \tag{7}$$

For simplicity, we consider the case where the input random variables are discrete, taking finite value from a set $\mathcal{X}$, then $p_\theta(i_1, \ldots, i_N)$ can be written as

$$p_\theta(i_1, \ldots, i_N) = \sum_{(x_1, \ldots, x_N) \in \mathcal{X}^N} \prod_{k=1}^N p_\theta(y_k = i_k|x_k) p(x_1, \ldots, x_N) \tag{8}$$

To show that $\bar{p}_\theta(i_N, \ldots, i_1)$ is an empirical approximation of $p_\theta(i_1, \ldots, i_N)$, it suffices to show that

$$\bar{p}_\theta(i_1, \ldots, i_N) = \sum_{(x_1, \ldots, x_N) \in \mathcal{X}^N} \prod_{k=1}^N p_\theta(y_k = i_k|x_k) \hat{p}(x_1, \ldots, x_N) \tag{9}$$

where $\hat{p}(x_1, \ldots, x_N)$ is the empirical frequency of the $N$-tuple $(x_1, \ldots, x_N)$ in the dataset $\{(x_1^n, \ldots, x_{T_n}^n) : n = 1, \ldots, M\}$. The result follows in a straightforward manner from the definition of $\bar{p}_\theta(i_1, \ldots, i_N)$:

$$\bar{p}_\theta(i_1, \ldots, i_N) = \frac{1}{T} \sum_{n=1}^M \sum_{t=1}^{T_n} \prod_{k=0}^{N-1} p_\theta(y_{t-k}^n = i_{N-k}|x_{t-k}^n)$$

$$= \frac{1}{T} \sum_{(x_1,\ldots,x_N)\in\mathcal{X}^N} \prod_{k=0}^{N-1} p_\theta(y_{t-k}^n = i_{N-k}|x_{t-k}^n = x_{N-k}) \times n(x_1,\ldots,x_N)$$

$$= \sum_{(x_1,\ldots,x_N)\in\mathcal{X}^N} \prod_{k=0}^{N-1} p_\theta(y_{t-k}^n = i_{N-k}|x_{t-k}^n = x_{N-k}) \times \frac{n(x_1,\ldots,x_N)}{T} \quad (10)$$

where $n(x_1,\ldots,x_N)$ denotes the number of $N$-tuple $(x_1,\ldots,x_N)$ in the dataset $\{(x_1^n,\ldots,x_{T_n}^n) : n = 1,\ldots,M\}$. The second equality is simply re-organizing the summation in the first expression according to the value of $(x_1,\ldots,x_N)$, i.e., accumulating all the terms inside the double-summation with the same value of $(x_1,\ldots,x_N)$ together. Further note that $p_\theta(y_{t-k}^n = i_{N-k}|x_{t-k}^n = x_{N-k})$ is independent of $t$ and $n$ for any given values of $i_{N-k}$ and $x_{N-k}$, so that

$$\prod_{k=0}^{N-1} p_\theta(y_{t-k}^n = i_{N-k}|x_{t-k}^n = x_{N-k}) = \prod_{k=1}^{N} p_\theta(y_k = i_k|x_k) \quad (11)$$

Then, we can conclude the proof by recognizing that $\hat{p}(x_1,\ldots,x_N) = n(x_1,\ldots,x_N)/T$.

## C   Optimizing Empirical-ODM by SGD is intrinsically biased

In this section, we show that the stochastic gradient of Empirical-ODM is intrinsically biased. To see this, we can express the (full batch) gradient of $\mathcal{J}(\theta)$ as

$$\nabla_\theta \mathcal{J}(\theta) = -\sum_{i_1,\ldots,i_N} p_{\text{LM}}(i_1,\ldots,i_N) \frac{\frac{1}{T}\sum_{n=1}^{M}\sum_{t}^{T_n} \nabla_\theta\left(\prod_{k=0}^{N-1} p_\theta(y_{t-k}^n = i_{N-k}|x_{t-k}^n)\right)}{\frac{1}{T}\sum_{n=1}^{M}\sum_{t}^{T_n} \prod_{k=0}^{N-1} p_\theta(y_{t-k}^n = i_{N-k}|x_{t-k}^n)} \quad (12)$$

Note that the gradient expression has sample averages in both the numerator and denominator. Therefore, full batch gradient method is less scalable as it needs to go over the entire training set to compute $\nabla_\theta \mathcal{J}(\theta)$ at each update. To apply SGD, we may obtain an unbiased estimate of it by sampling the numerator with a single component while keeping the denominator the same:

$$-\sum_{i_1,\ldots,i_N} p_{\text{LM}}(i_1,\ldots,i_N) \frac{\nabla_\theta\left(\prod_{k=0}^{N-1} p_\theta(y_{t-k}^n = i_{N-k}|x_{t-k}^n)\right)}{\frac{1}{T}\sum_{n=1}^{M}\sum_{t}^{T_n}\prod_{k=0}^{N-1} p_\theta(y_{t-k}^n = i_{N-k}|x_{t-k}^n)}$$

However, this implementation is still not scalable as it needs to average over the entire training set at each update to compute the denominator. On the other hand, if we sample both the numerator and the denominator, i.e.,

$$-\sum_{i_1,\ldots,i_N} p_{\text{LM}}(i_1,\ldots,i_N) \frac{\nabla_\theta\left(\prod_{k=0}^{N-1} p_\theta(y_{t-k}^n = i_{N-k}|x_{t-k}^n)\right)}{\prod_{k=0}^{N-1} p_\theta(y_{t-k}^n = i_{N-k}|x_{t-k}^n)}$$

then it will be a biased estimate of the gradient (12). Our experiments in Section 4 showed that this biased SGD does not perform well on the unsupervised learning problem.

## D   Gradient formula for SPDG

In this section, we derive the gradient formula for the SPDG algorithm in Algorithm 1. We first derive the formula for $\frac{\partial L_t^n}{\partial \nu_{i_1,\ldots,i_N}}$:

$$\frac{\partial L_t^n}{\partial \nu_{i_1,\ldots,i_N}} = p_{\text{LM}}(i_1,\ldots,i_N) \prod_{k=0}^{N-1} p_\theta(y_{t-k}^n = i_{N-1}|x_{t-k}^n) \quad (13)$$

Then, we derive the gradient formula for $\frac{\partial}{\partial \nu_{i_1,\ldots,i_N}} \sum_{i_1,\ldots,i_N} p_{\text{LM}}(i_1,\ldots,i_N) \ln(-\nu_{i_1,\ldots,i_N})$:

$$\frac{\partial}{\partial \nu_{i_1,\ldots,i_N}} \sum_{i_1,\ldots,i_N} p_{\text{LM}}(i_1,\ldots,i_N) \ln(-\nu_{i_1,\ldots,i_N}) = -\frac{p_{\text{LM}}(i_1,\ldots,i_N)}{\nu_{i_1,\ldots,i_N}} \quad (14)$$

Finally, we derive the gradient formula for $\frac{\partial L_t^n}{\partial \theta}$:

$$\frac{\partial L_t^n}{\partial \theta} = \sum_{i_1,\ldots,i_N} p_{\text{LM}}(i_1,\ldots,i_N)\nu_{i_1,\ldots,i_N}$$

$$\times \sum_{m=0}^{N-1} \frac{\partial}{\partial \theta} p_\theta(y_{t-m}^n = i_{N-m}|x_{t-m}^n) \prod_{k=0,k\neq m}^{N-1} p_\theta(y_{t-k}^n = i_{N-k}|x_{t-k}^n) \qquad (15)$$

where the gradient term $\partial p_\theta/\partial \theta$ depends on the specific model for the classifier $p_\theta(y|x)$ and can be calculated easily by substituting its expression.

# E    Experiment Details

In the experiment, we implement the model with Python 2.7 and Tensorflow 0.12.

In training of models both on OCR and Spell-Corr task, we initialize the linear model's parameters (primal variable) with $w_{init} = 1/dim(x)$ and $\gamma = 10$, where $dim(x)$ is the dimension of input. And we initialize the dual parameters $V_{init}$ with uniformly distributed random variables $v \sim U(-1, 0)$. We set the learning rate for primal parameter $\mu_\theta = 10^{-6}$ and learning rate for dual parameter $\mu_v = 10^{-4}$. We use Adam optimization to train our model.

The test set of OCR is generated also from UWIII database, but avoiding overlap with training set. The size of test set of OCR is 15000. Furthermore, the size of the test set of Spell-Corr is also 15000 without overlapping with the training set.

# F    The details of visualizing the high-dimensional cost functions

Since $\mathcal{J}(\theta)$ is a high-dimensional function, it is hard to visualize its full profile. Instead, we use the following procedure to partially visualize $\mathcal{J}(\theta)$. First, since the supervised learning of linear classifiers is a convex optimization problem, from which we could obtain its global optimal solution $\theta^\star$.[9] Then, we randomly generate two parameter vectors $\theta_1$ and $\theta_2$ and plot the two-dimensional function $\mathcal{J}\big(\theta^\star + \lambda_1(\theta_1 - \theta^\star) + \lambda_2(\theta_2 - \theta^\star)\big)$ with respect to $\lambda_1, \lambda_2 \in \mathbb{R}$, which is a slice of the cost function on a two-dimensional plane.

For the profile of $\mathcal{L}(\theta, V)$ in (5), similar to the case of $\mathcal{J}(\theta)$, in order to visualize $\mathcal{L}(\theta, V)$, we first solve the supervised learning problem to get $\theta^\star$. Then we substitute $\theta^\star$ into (5) and maximize $\mathcal{L}(\theta^\star, V)$ over $V$ to obtain $V^\star = \{\nu_{i_1,\ldots,i_N}^\star\}$, where $\nu_{i_1,\ldots,i_N}^\star = -1/\frac{1}{T}\sum_{n=1}^{M}\sum_{t=1}^{T_n}\prod_{k=0}^{N-1} p_{\theta^\star}(y_{t-k}^n = i_{N-k}|x_{t-k}^n)$. We also randomly generate a $(\theta_1, V_1)$ (with the elements of $V_1$ being negative) and plot in Figure 2a the values of $\mathcal{L}(\theta^\star + \lambda_p(\theta_1 - \theta^\star), V^\star + \lambda_d(V_1 - V^\star))$ for different $\lambda_p, \lambda_d \in \mathbb{R}$. Clearly, the optimal solution (red dot) is at the saddle point of the profile.

# G    Additional visualization of $\mathcal{J}(\theta)$

In Figures 3, 4 and 5, we show three visualization examples of $\mathcal{J}(\theta)$ for the OCR dataset on three different affine spaces, part of the first example was included in Figure 1. The six sub-figures in each example show the same profile from six different angles, spinning clock-wise from (a)-(f). The red dots indicate the global minimum.

In Figure 6, we show the same type of profiles as above except using synthetic data for of a binary classification problem. First, we sequentially generated a sequence of states from $0, 1$ by an hidden Markov model. Then we sample the corresponding data points from two separate 2-dimensional Gaussian models. accordingly.

(a)  (b)  (c)

(d)  (e)  (f)

Figure 3: Profile Example I: $\mathcal{J}(\theta)$ for the OCR dataset on a two-dimensional affine space

(a)  (b)  (c)

(d)  (e)  (f)

Figure 4: Profile Example II: $\mathcal{J}(\theta)$ for the OCR dataset on a two-dimensional affine space

# H  Additional visualization of $\mathcal{L}(\theta, V)$

Figure 7 shows the profile of $\mathcal{L}(\theta, V)$ for the OCR data set on a two-dimensional affine space viewed from nine different angles. The red dots show the saddle points of the profile, one for each angle.

Figure 5: Profile Example III: $\mathcal{J}(\theta)$ for the OCR dataset on a two-dimensional affine space

Figure 6: Complete $\mathcal{J}(\theta)$ profile created from 2-dim synthetic data with two parameters

Figure 7: Profile of $\mathcal{L}(\theta, V)$ for the OCR dataset on a two-dimensional affine space. Red dots show the saddle points (the optimal solution) of the profile from nine different angles.

## Footnotes

[9]Note that, we solve the supervised learning only for the purpose of understanding our proposed unsupervised learning cost $\mathcal{J}(\theta)$. In our implementation of the unsupervised learning algorithm, we do not use any of the training label information nor supervised learning algorithms.