[Reviews · NeurIPS 2017]

Reviewer 1



This paper proposes an algorithm to learn a sequence classifier without labeled data by using sequential output statistics (language model). The proposed formulation is hard to optimize in its functional form and a stochastic primal dual gradient method is developed to efficiently solve this problem. Compared to earlier work this formulation is less inclined to get stuck at a trivial solution and doesn’t require a generative model. Experimental results on two real world datasets show that the proposed method results in lower error rate compared to its base line methods. Here are few comments. 1. In figure 2(a), what does the two axis represent (lambda_d and lambda_p)? 2. The basic idea of the paper and its primal dual stochastic gradient formulation seems convincing. It would be nice however, to include other baseline comparisons, namely from [11] and [30]. Also it would be convincing to add more datasets. 3. It would be nice to provide explicit gradient formulas (step 5 of algorithm 1).

Reviewer 2



This paper studies the problem of learning a sequence classifier without labeled data by using sequential output statistics. This is motivated by applications where labels are costly to obtain, while sequential output statistics can be obtained easily. One typical example is OCR, where the output statistics are the language models. It proposes an unsupervised learning cost function, which is intuitive: view the mapping from the input sequence to the output sequence as transforming the input distribution to a distribution in the output domain, and the cost is the cross entropy between the output statistics and the transformed distribution. The authors discuss some nice properties of the cost: though it's non-convex, compared to existing works it has the property of coverage seeking that is better for optimization. It also transforms the cost to its primal-dual form and proposes a stochastic gradient method. Experiments on two real applications show that this approach works well (gets results close to that of supervised learning), while the alternatives fail miserably. The presentation is very clear. Detailed comparisons are given to related works, though I'm not familiar enough with this direction to see if it covers all. Overall, the proposed approach is intuitive and the paper gives detailed discussions about advantages over existing works. Experimental results provide strong support for the effectiveness of the method. minor: --Line 283: rate rate --> rate

Reviewer 3



This paper presents a method for predicting a sequence of labels without a labeled training set. This is done by incorporating a prior probability on the possible output sequnces of labels, that is a linear classifier is sought such that the distribution of sequnces it predicts is close (in the KL sense) to a given prior. The authors present a cost function, an algorithm to optimize it, and experimental results on real data of predicting characters in two tasks: OCR and spell correction. The paper is technically sound and generally clear. The idea is novel (some prior work is cited by the authors) and interesting. The main reason for my score is a concern regarding the difficulty of obtaining a useful prior model p_LM. Such models usually suffer from sparseness of data, that is unless the vocabulary is quite small, it is common for some test samples to have zero probability under the prior/LM learned from a training set. It is true that this isuue is well known and solutions exist in the literature, however I would like the authors to adress the influence of it on their method. Another aspect of the same issue is the scaling of the algorithm to large vocabularies and/or long sub-sequences (parameter N in the paper). Indeed, the experiments use quite a small vocabulary (29 characters) and short sub-sequences (Ngrams with N=2,3). In NLP data at word level, the vocabulary is orderes of magnitude larger. Some more comments: - I would also like the authors to adress the application of their method to non-NLP domains. - The inclusion of the supervised solution in the figures of the 2D cost raises a question - it seems the red dot (the supervised solution) is always in a local minimum of the unsupervised cost - I don't understand why this should be the case. - If it is assumed that there is a structure in the output space of sequences, why not incorporate this structure into the classifier, that is why use a point prediction p(y_t|x_t) of a single label independently of its neighbors? Specific comments: (*) lines 117-118, 'negative cross entropy', isn't the following formula the cross entropy (and not the negative of it)? (*) line 119, this sentence is somewhat confusing, if I understand correctly \bar p is the expected frequency of a *given* sequence {i_1,...i_n}, and not of *all* sequences. (*) lines 163-166, here too the sign of the cross entropy seems to be reversed, also when referred to in the text, e.g in line 163, if p_LM goes to 0, then -log(0) is +inf and not -inf as written in the paper, and it indeed suits the sentence in line 167, since you expect that to penalize a minimization problem, some term goes to +inf and not to -inf. Typos: (*) lines 173 and 174, "that p_LM" seems wrong, did the authors mean "for which p_LM" ? (*) line 283, word 'rate' appears twice.